# Learning Causal Discovery

## Abstract

Causal discovery (CD) from time-varying data is important in neuroscience, medicine, and machine learning. Techniques for CD include randomized experiments which are generally unbiased but expensive. It also includes algorithms like regression, matching, and Granger causality, which are only correct under strong assumptions made by human designers. However, as we found in other areas of machine learning, humans are usually not quite right and human expertise is usually outperformed by data-driven approaches. Here we test if we can improve causal discovery in a data-driven way. We take a perturbable system with a large number of causal components (transistors), the MOS 6502 processor, acquire the causal ground truth, and learn the causal discovery procedure represented as a neural network. We find that this procedure far outperforms human-designed causal discovery procedures, such as Mutual Information, LiNGAM, and Granger Causality both on MOS 6502 processor and the NetSim dataset which simulates functional magnetic resonance imaging (fMRI) results. We argue that the causality field should consider, where possible, a supervised approach, where CD procedures are learned from large datasets with known causal relations instead of being designed by a human specialist. Our findings promise a new approach toward improving CD in neural and medical data and for the broader machine learning community.

## 1 Introduction

Decision relevant insights are usually about causation. In public health, practitioners want to decide policy interventions based on causal relations (Glass et al., 2013). In sociological research, scientists want to use causal language to answer what effect a specific behavior can have (Sobel, 1995). In biomedical science, researchers ask what causal mechanism a new medicine has (Imbens & Rubin, 2015). In neuroscience, we may ask about the causal role of a neuron's activation or a lesion in a brain region (Marinescu et al., 2018). Causal insights are at the heart of science, engineering, and medicine.

To uncover the causal relationships inside a complex system, researchers often start by estimating the causal influence from one element to another. Randomized controlled trials (RCT) is regarded as the gold standard in clinical and societal problems (Zheng et al., 2020; Sherman et al., 2005). However, conducting an RCT is expensive even without strict standards of comparable group settings, and faces many ethical restrictions. For example, it is unethical to make mothers to ask how that affects birth weights. RCTs are a clean way of establishing causality but are often unethical or not feasible, and therefore we need other ways to estimate causality from other data sources.

Revealing causal relationships from observational data, known as *causal discovery*, is thus an active field of causality research. Some causal discovery methods, such as constraint-based methods, use conditional independence relationships. Classical constraint-based causal discovery algorithms include PC (Spirtes et al., 2000b), FCI (Spirtes et al., 2000a). While these algorithms show us the possibility of inferring causality from all kinds of purely observational data, they are only valid when the faithfulness assumption, justified by human experts, holds. Such methods are also somewhat limited in terms of power and only provide an equivalence class of possible causal models, the Markov Equivalence Class. Other causal discovery methods are based on the Functional Causal Model (FCM), which depicts the directional influence among the elements in a system by a set of equations. Methods such as LiNGAM (Shimizu et al., 2006), ANM (Hoyer et al., 2008), PNL (Zhang & Hyvärinen, 2010) try to recognize the role of cause by looking for noise-caused asymmetric

effects between cause and effect. These methods derive from assumptions about noise terms, again justified by human experts. Other causal discovery methods use scores like Bayesian Information Criterion (BIC) and Generalized Score (Chickering, 2002; Huang et al., 2018; Zheng et al., 2018; Zhu & Chen, 2019; Goudet et al., 2017). Yet another causal discovery method uses temporal prediction ideas and the assumption that cause always precedes effect, e.g. Granger Causality (Granger, 1969). Causal discovery from observational data (see https://www.cclear.cc) is now a popular field within machine learning and statistics (Pearl, 2010; Schölkopf et al., 2021; Glymour et al., 2019).

Nonetheless, all these methods depart from strong human assumptions about causality (see Figure 1). The causal discovery methods implement human intuition in a different form to discover causality from the dataset at hand. If these assumptions are correct, then we can use domain knowledge to construct good inference algorithms. If they are not, we have to construct algorithms that we expect to be a good approximation. However, human intuition of what constitutes a good approximation may be limited. Besides, if the history of the machine learning field is any indication, humans often do not have the right or at least complete intuition (Sutton, 2019). Therefore, we need to ask if the same lesson that learning is better than human ingenuity can be replicated in the causality domain.

Machine Learning (ML) has shown rapid progress in recent decades and it has been successfully applied to many domains such as text generation, image classification, and decision making (Shinde & Shah, 2018). Different from causal discovery methods, ML nurtures algorithms in a data-driven way (see Figure 1), with only weak human assumptions. These techniques outperform human operation in many fields, such as face recognition (Lu & Tang, 2015), Go (Silver et al., 2016). For causal discovery we want a way of generating causal estimators without hand-engineering in a data-driven way. If we had causal ground truth data, we may expect that the strength of modern machine learning, including neural networks, may allow efficient learning of causal discovery.

Here we use supervised learning on a large dataset of known causal relations to discover an algorithm that will correctly identify causality from unsupervised observations only. We take a deterministic complex system with many causal components, the microprocessor MOS 6502. We first conduct a perturbation analysis of all unique transistors to produce ground-truth causality relationships. We thus know which transistors affect which other transistors. The problem of causal discovery from time series here boils down to deciding if one element causally influences the other. It is a mapping from a two-by-time vector (both transistors by time) into a binary outcome, causal or not. To implement this mapping, we can use the obvious sequence encoders like long short term memory (LSTM), temporal convolutional network (TCN), and Transformer. We thus construct a system that can learn the causal discovery procedure.

Our contribution can be concluded as follows:

- We use learning to generate a new causal discovery procedure and show that it works better than algorithms based on human intuition.

- We examine our learning way under different levels of noise and its internal invariance within different behaviors, showing its robust anti-noise ability and stable generalization across different behaviors executed by the same system.

- We conduct an explanation study by Grad-CAM on the attention layer of our Transformer-based architecture and find that our procedure has helped the models learn causal features.

## 2  Methods

To be able to test algorithms of causal discovery, we need to have a database of known causal relations. We therefore use a deterministic system that contains many causal relations, the MOS 6502 Microprocessor. We can readily measure causal effects by perturbing transistors and seeing how this affects voltages a short period of time later (one half-clock). We can then ask if an algorithm that is trained on a subset of whole transistors can infer causal influences inside another subset that is different from the training set.

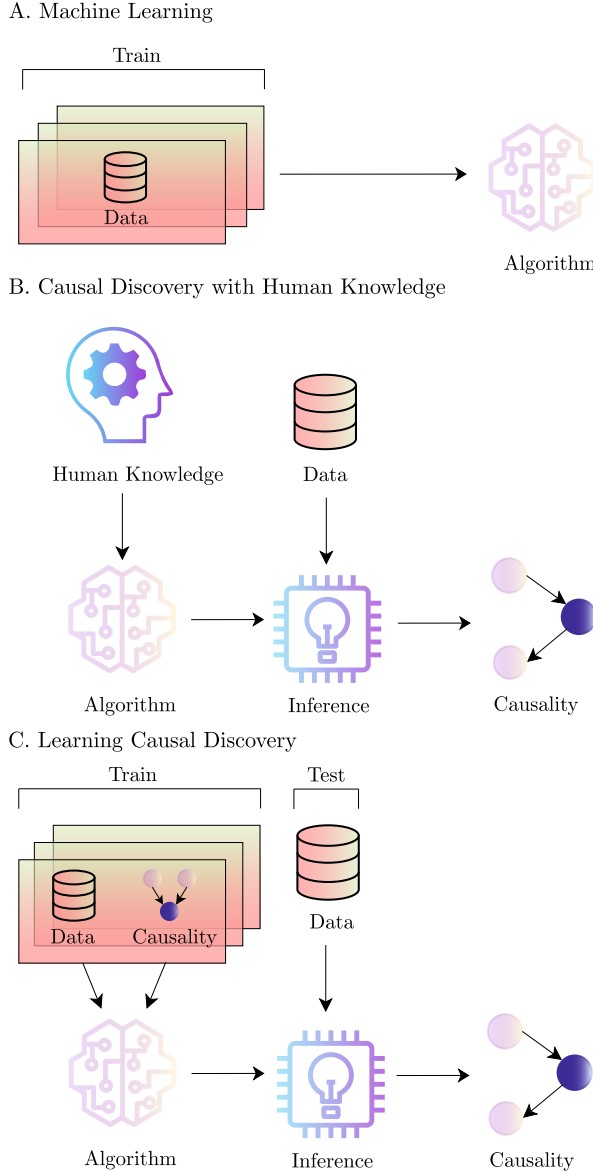

Figure 1: **A.** In machine learning, algorithms (predictors) are generated from data and then used in inference on unseen data. **B.** In traditional causal discovery procedures, algorithms are not generated from data but from human knowledge. These algorithms infer causality in unseen data by fitting unseen data into human assumptions. **C.** Similar to machine learning, learning causal discovery generates algorithms (predictors) from lots of observational data and known causality, expecting them to learn causality with weak human assumptions, in a data-driven way. Then trained algorithms (predictors) are used to infer causality in unseen data.

To be able to know about causality, we need to start with the perturbation experiment of the MOS 6502. We take a C++ optimized MOS 6502 simulator (Jonas & Kording, 2017) with three game recordings (Donkey Kong, Pitfall, Space Invaders) as the target system. We first acquire the causal relationship among transistors by single element perturbation analysis and define it as the ground truth of the causality inside the MOS 6502

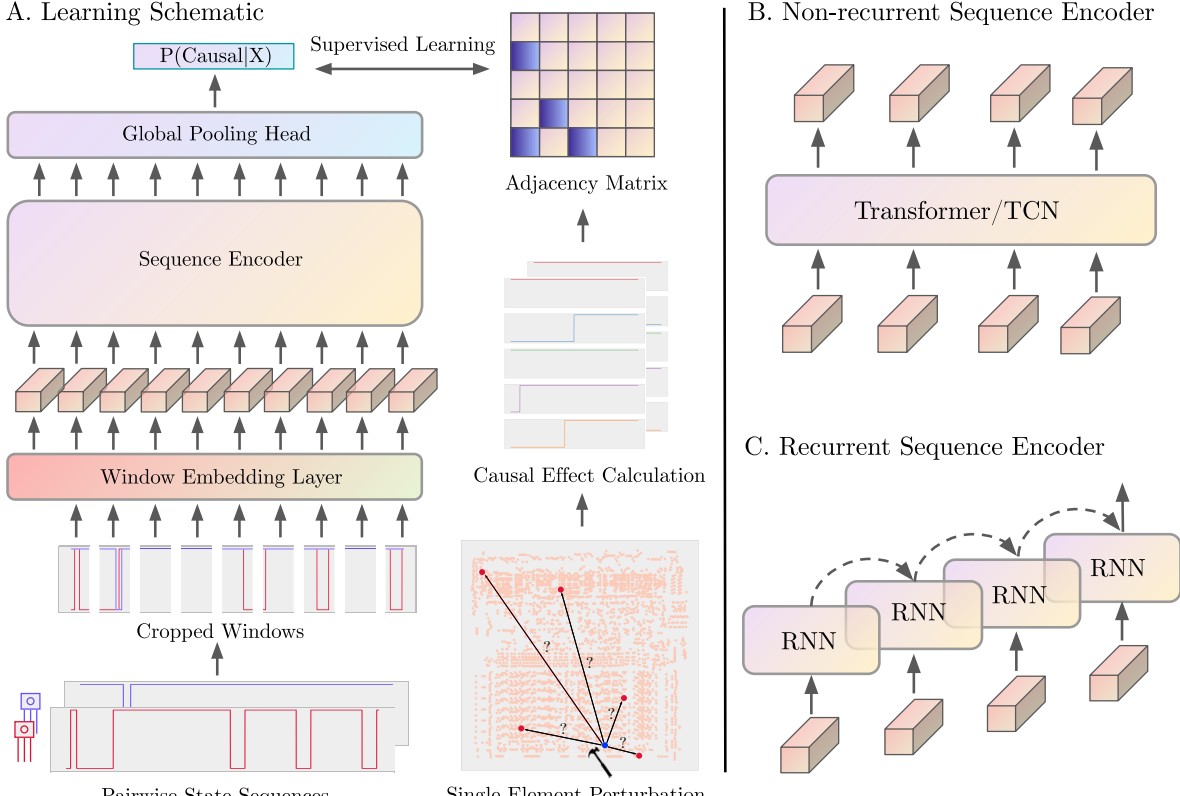

Figure 2: Methods for learning causal discovery. **A.** Steps in the learning schematic: On the left is the causal discovery model structure, in which the sequence encoder could be a non-recurrent or recurrent sequence encoder. On the right is the perturbation workflow used to get causal ground truth inside the microprocessor. **B.** Non-recurrent sequence encoders process all elements in the sequence at the same time, such as Transformer and TCN. **C.** Consisting of recurrent neural network units, non-recurrent sequence encoders process every element in the sequence order and transit state one by one, such as RNN, LSTM, and GRU.

system, which is utilized as a supervised signal and a validation standard in our causal discovery procedure (we could have used the netlist but that one has no directions distinguishing cause and effect). This way, we know the ground truth causal influence of transistors upon one another, which we can use for learning a strategy and for testing.

## 2.1 Single Element Perturbation Analysis

Identifying the causal relationships in MOS 6502 only from observations is hard. MOS 6502 is a large complex system consisting of 3510 transistors and 1904 connection elements, which allows a multi-input and multi-output (MIMO) to form among transistors. Even though the existing netlist provides us with what connection elements look like, lacking directional information and what paths it flows through the MIMO connections, it is tough to give conclusions about what cause-effect one transistor has on the others. To uncover the causal relationships among transistors, we conduct a single element perturbation experiment, an approach often used to infer causality in the brain network, cellular events, and genetics (Paus, 2005; Meinshausen et al., 2016; Welf & Danuser, 2014). The main idea here is that when the cause element is perturbed, its downstream targets will be affected. By calculating the difference when being perturbed and not, we can determine if a causal influence flows from cause to effect.

Temporal precision is crucial for capturing causality in time-varying data. Low temporal precision might hide the temporal difference between cause and change at the same time point, making the cause and effect unidentifiable. Therefore, although the half-clock-wise recording rule of transistor state in the original version could highly reduce the number of samples and redundancy within the clock, it often regards the change of cause and effect happening at the same time point, leading to more confusion in discriminating between the cause and effect. Therefore, we adopt the step-wise recording rule rather than the original half-clock recording rule, which gives a much higher temporal resolution of the transistor state and a better view of how causality happens. To be more specific, we define the update iteration of a batch of the simultaneous nets as one step, whose update times vary in different clocks. We simulate the clock signal function in the digital circuits to make steps in different half-clocks the same length by padding the last collected step in each half-clock. Here we set the step limitation to be 2000, so for a 256 half-clock runtime, the total sequence length of a transistor would be 512000 steps.

High repeatability is a regular but tricky phenomenon that happens in the ideal system. Because of MIMO connections, it is common for some transistors to have almost the same state sequence in a short period. To reduce redundancy in the perturbation experiment and causal discovery procedure, we collect the first 256 half-clock state sequences of all transistors and only retain the transistors with unique sequences. The following perturbation experiment and discovery procedure are all based on this set of transistors, whose quantity somewhat varies across games, but generally is around 800. We thus only analyze transistors with unique behavior.

In a complex system, cause-effect may be continuously spread to more and more indirect targets as time goes forward. To simplify this kind of influence and focus on the relatively direct cause-effect relations, we limit the causal effect calculation only to the first half-clock that the perturbation of the cause transistor actually begins working, where the transistor state sequence under regular operation first changes.

Based on the above, we conduct a single element perturbation on every unique transistor, where every time we impose the perturbation on a single transistor and measure the causal effect it has on the other transistors, and repeat it for every transistor individually. A significant advantage of performing such perturbation is that it gives us the ability to reject all the confounding effects brought on by predecessors. To be more specific, we force the state of an individual transistor to be ON in the microprocessor runtime if this transistor is supposed to be OFF in normal operation, and vice versa (Algorithm 2.1, line 3). Since the step number in each half-clock is not consistent, we record the markers to label when a half-clock starts and ends. We then use markers to do padding to reconstruct the equal interval sequence. In order to speed up simulation and save storage costs, we simulate the same length as regular operation but only recorded the perturbation result in the first clock that the cause transistor starts to change. We record the regular runtime for 256 half-clocks for all transistors and repeated this procedure for three games, respectively. Denote that $\boldsymbol{x}_i \in \mathbb{R}^L$ is the transistor state sequence of perturbed transistor $i$ and $\boldsymbol{x}_j \in \mathbb{R}^L$ is the other transistor sequence of transistor $j$. We define that $\boldsymbol{x}_{j,do(\boldsymbol{x}_i=p)}$ is the state sequence of transistor $j$ when imposing perturbation on the transistor $i$, and the Average Treatment Effect (ATE) of transistor $i$ on transistor $j$ in one half-clock $ATE_{i,j}$ is defined by the expectation of the difference between the regular and the perturbation condition (Algorithm 2.1, line 7).

$$ATE_{i,j} = \mathbb{E}[\boldsymbol{x}_{j,do(\boldsymbol{x}_i=p)} - \boldsymbol{x}_j] \tag{1}$$

We then convert ATE of all the transistors to a binary adjacency matrix $\boldsymbol{A}$ to describe the relationship between any pair of transistors as the equation below. Here the transistor $i$ is the cause of transistor $j$ if the average treatment effect $ATE_{i,j}$ is not zero, while for reversal causal and unrelated relationships, the average treatment effect should be zero (Algorithm 2.1, line 8):

$$\boldsymbol{A}_{i,j} = \mathbf{1}_{\text{ATE}_{i,j}>0} \tag{2}$$

---

**Algorithm 1** Adjacency Matrix Generation

---

1: Start with $N$ observed elements inside system
2: **for** $i = 0, 1, ..., N$ **do**
3:     Perturb $element_i$ and record all elements' states $\{\boldsymbol{x}_0, \boldsymbol{x}_1, ..., \boldsymbol{x}_N\}$
4: **end for**
5: **for** $i = 0, 1, ..., N$ **do**
6:     **for** $j = 0, 1, ..., N$ **do**
7:         Calculate average treatment effect $ATE_{i,j}$ from $element_i$ to $element_j$ across time as equation 1
8:         Calculate the entry $\boldsymbol{A}_{i,j}$ of adjacency matrix as equation 2
9:     **end for**
10: **end for**

---

## 2.2 Learning Procedure

In this subsection, we define the causal discovery procedure based on deep learning in two steps, data preprocessing and baseline setting. Here we focus on the causal discovery in the game Donkey Kong and utilize its simulation data acquired from single element perturbation experiment, including the regular runtime state sequence $\boldsymbol{x}$ and the adjacency matrix $\boldsymbol{A}$. Note that in the causal discovery procedure, we aim at exploring the potential of inferring pairwise causality from observational data without the direct view of any perturbation. Therefore, the adjacency matrix $\boldsymbol{A}$ is only adopted as the label to do supervised learning and algorithm validation. After data preprocessing, the processed state sequences of transistor pairs are fed into our discovery architecture.

---

**Algorithm 2** Learning Procedure

---

**Input:** training set $\mathbb{T} = \{(\boldsymbol{X}_n, \boldsymbol{A}_n)\}_{n=1}^{N_{data}}$, a dataset of pairwise element state sequences; sample $\boldsymbol{X}_n = (\boldsymbol{x}_i \parallel \boldsymbol{x}_j)$, the stack of pairwise state sequences of $element_i$ and $element_j$, $\boldsymbol{A}_n$ is the entry $\boldsymbol{A}_{i,j}$ of adjacency matrix.
**Input:** $\theta$, initial estimator parameters
**Output:** $\hat{\theta}$, the trained causality estimator
**Hyperparameters:** $N_{epochs} \in \mathbb{N}, \eta \in (0, \infty)$,
1: **for** $i = 1, 2, ..., N_{epochs}$ **do**
2:     **for** $n = 1, 2, ..., N_{data}$ **do**
3:         $P \leftarrow Estimator(Causal|\boldsymbol{X}_n, \theta)$
4:         $L(\theta) = -(\boldsymbol{A}_n \cdot \log(P) + (1 - \boldsymbol{A}_n) \cdot \log(1 - P))$
5:         $\theta \leftarrow \theta - \eta \cdot \nabla L(\theta)$
6:     **end for**
7: **end for**
8: **return** $\hat{\theta} = \theta$

---

### 2.2.1 Data Preprocessing

After acquiring clean sequence data and binary labels, we first divide all unique transistors into three sets: training, validation, and testing in the ratio $6 : 2 : 2$. We divide in such a way, that no transistor shows up in two of the three sets to minimize the risk of leakage. We then construct transistor pairs inside each set independently, regarding each pairwise relationship as a data sample, constructing 653672 samples in total with 9844 positive samples. For instance, even though transistor $i$ and transistor $j$, and transistor $j$ and transistor $i$ are the same sequences, they were different data samples (Algorithm 2: Input). We repeat this splitting 5 times with different seeds to guarantee the fairness of our results. Compared to non-causal relationships, causal pairs are in the minority, leading to a severe bias towards the non-causal side. In order to mitigate the negative effect of the sample imbalance, positive samples are randomly over-sampled to reach the same quantity as negative samples in the training set. In reality, causal relationships are always sparse compared to non-causal relationships, which become

more severe when more elements are involved in a system. Therefore, we kept the validation and test sets unchanged to reveal what happened in reality and better evaluate the discovery quality of our procedure.

### 2.2.2 Architecture

Here we illustrate the architecture of learning causal discovery. Note that the high-resolution state recording is very sparse and long, which makes direct feeding in the network less informative and hard to compute. If we take a zoom-in view of our state sequences in the microprocessor system, we will see that there are a lot of constant states and relatively fewer turning points (see Figure 4). Therefore, we need the model to discriminate and capture the informative periods where cause-effect relations may be visible. The system must also be able to accommodate long input sequences. Inspired by Dosovitskiy et al. (2020), we convert the pairwise input $\boldsymbol{X} \in \mathbb{R}^{L \times 2}$ into a sequence of window embeddings $\boldsymbol{X}_w \in \mathbb{R}^{N \times C}$ by the window embedding layer. Here we use a 1D convolutional layer as the window embedding layer since such works like Text-CNN (Kim, 2014), TCN (Lea et al., 2016) had proved that convolutional kernels are effective in exacting sequence information. Benefiting from the sharing parameters of convolutional kernels, we encode windows in a computation-efficient way. We use the window embedding layer to capture the causal features inside an individual window, such as the temporal lag of effect in a short period. We also add a [*class*] token as the first window embedding and encode position information for each window. In order to capture the information across windows and aggregate the window features together, the window embeddings are then fed into a sequence encoder, which could be a non-recurrent sequence model like Transformer (Vaswani et al., 2017) and TCN (Lea et al., 2016), or recurrent sequence model like LSTM (Hochreiter & Schmidhuber, 1997), GRU (Cho et al., 2014). The output of the sequence encoder is globally average pooled and projected to the probability of binary classification by a linear layer (Algorithm 2, line 3-5). We thus use a relatively standard architecture from the time series or natural language field.

### 2.2.3 Baseline Setting

- Input: We set sequence length $L$ to be 51200 by resampling with a fixed 10 time steps as the interval from the original 256 half-clock sequences. The window embedding sequence length $N$ is 101, consisting of one [*class*] token and 100 window embeddings when set window size to be 512. The number of window embedding channels $C$ is 128.

- Encoder: We use Bi-directional LSTM, TCN, and Transformer as sequence encoders. For LSTM and TCN, we use layer depth 2 with 128 as the hidden size. For Transformer, we use layer depth 4 with hidden size 128 and 8 attention heads.

- Optimization: Pooler output is regarded as $P(Causal|\boldsymbol{X})$ and compared with the adjacency matrix to acquire cross-entropy loss. All models are optimized with AdamW with a learning rate 0.001 and batch size 256 for 50 epochs. Weight decay 0.05 and early stopping with patience 5 epochs are used to prevent over-fitting. The learning rate is warmed up for 5 epochs and adjusted by the Cosine Annealing scheduler in the rest process.

## 3 Empirical Study

Intending to uncover the causal effect inside the MOS 6502 and the robustness of our learning causal discovery procedure, we carry out multiple empirical studies, including a causal effect analysis, regular evaluation, noise tests, internal invariance assessment and explanation study in the context of MOS 6502. To test the generality of the approach we then also test on the NetSim (Smith et al., 2011) dataset of simulated fMRI results, a dataset we did not use when developing our algorithms.

### 3.1 Causal Effect

It is both an interesting and difficult question to determine how much causal influence there is and what role it plays in a complex system. Single element perturbation on each transistor in the game Donkey Kong

provides us with the causal effect of every individual transistor. Figure 3 shows the average treatment effect of three individual transistors, 1, 990 and 3057.

A. Average treatment effect of lesioning transistor 1

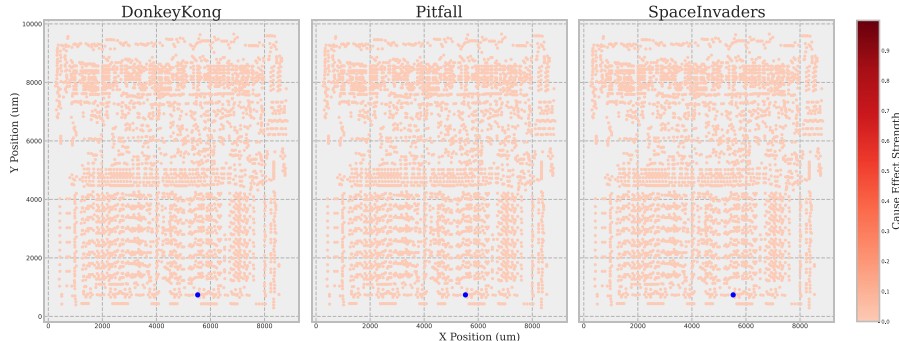

B. Average treatment effect of lesioning transistor 990

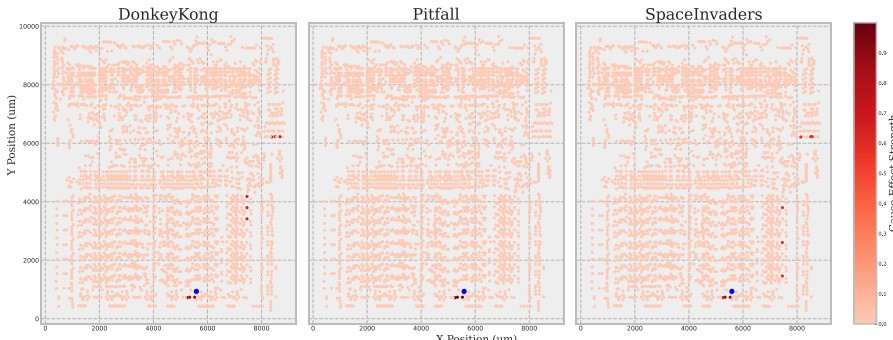

C. Average treatment effect of lesioning transistor 3057

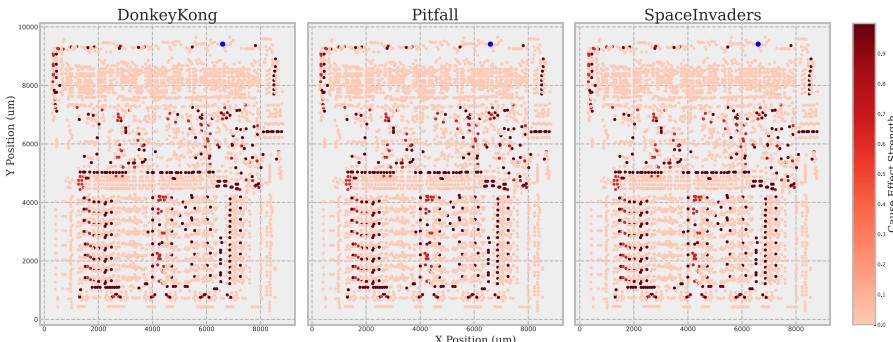

Figure 3: Causal effect of lesions inside MOS 6502 when executing different games (the blue point is the lesion transistor). **A.** Average treatment effect (ATE, measured during one half-clock, 2000 time steps) when stimulating transistor 1; **B.** ATE of transistor 990; **C.** ATE of transistor 3057

The first subplot shows an interesting phenomenon that the perturbation of transistor 1 does not cause differences to any other transistors. It indicates that even though the connected netlist guarantees that its perturbation output seems to be transferred to other elements, the following transistors can not be affected. This demonstrates that physical connection information is not enough to provide complete information about causal relationships.

In 3.B and C, the transistor 990 and 3057 exhibit different average treatment effects on the other transistors, which suggests variation in the causal relationship between the cause transistor and other transistors. As the clock transistor is connected to cclk wire, transistor 3057 has a stronger cause effect on the other transistors than transistor 990 in all games, indicating its more crucial role in the MOS 6502 system. In addition, we see that the effects of transistor 990 vary in different games, which violates the stability of causal relationships in different environments. We assume that this is induced by the variation of indirect effects since the causal effect here is defined by the lesioning effect. We see that the cause-effect in the MOS 6502 system is nontrivial and strongly associated with different functional areas. To be more specific, the distribution of causal effect strength is associated with their location, which indicates different circuit units that conduct different functions, such as clock registers and arithmetic-logic units. For instance, in controlling the chronological signal of various circuit units, the transistors consisting of a clock register have the largest total causal effect on the system. More importantly, most of the distribution of causal effects is similar in the other two games, Pitfall and Space Invaders, indicating that for a large complex system, even if the behaviors they are executing are different, the causal structure behind them might be highly similar. We regard this as a high-level function generalization in different behaviors, which demonstrates the robust stability of causality. Having defined causality in the microprocessor thus sets us up to check how well we can do causal discovery from time series.

## 3.2 Evaluation of classical and learned causal discovery processes

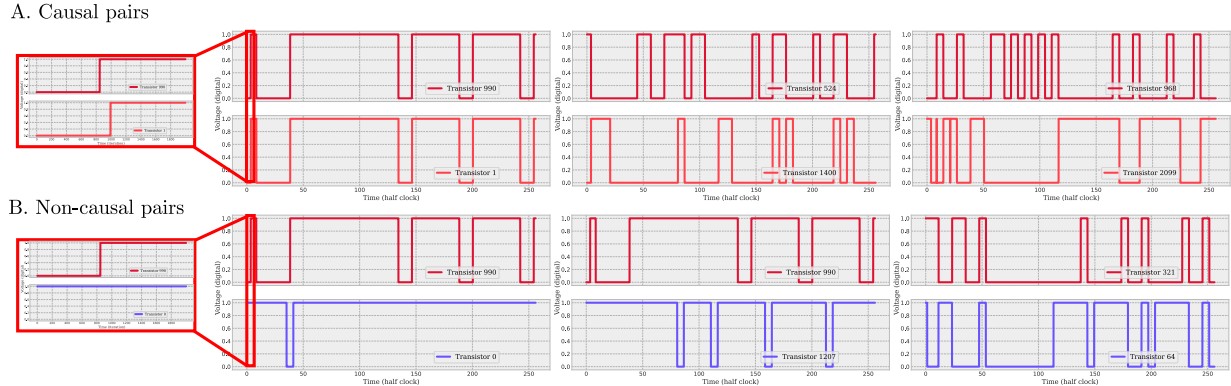

Figure 4: Transistor pair examples of different relationships. **A.** Causal pairs of transistors: 990 *to* 1, 524 *to* 1400 and 968 *to* 2099. **B.** Non-causal pairs of transistors: 990 *and* 0, 990 *and* 1207, and 321 *and* 64

Many aspects of time series may be indicative of causality. The left part of Figure 4 shows some examples of causal and non-causal pairs, uncovered by perturbation analysis. Taking a zoom-in view of transistors 990 and 1 shows that the causal feature here is the time lag between cause and effect in the turning point of cause. Even though it is the easiest sample of these three positive samples, the sparsity of the causal information is difficult for human-assumption methods to exact. In addition, the time lag is not consistent in the other samples, which violates the need for the fixed time lag in human-assumption methods such as Granger Causality (Granger, 1969) and Vector Autoregression models (Moneta et al., 2011). Even if cropping windows and multiple time lags are attempted to compensate for those limitations, various difficult scenarios like the other two positive samples are beyond human observation and assumptions, which might have potential specialized causal features in a specific kind of complex system. Many features may be usable to infer causality from the time series.

Here we evaluate how well learning discovery procedures do and compare them with traditional bi-variate relativity metrics: Pearson Correlation, Mutual Information, and Linear Granger Causality (Granger, 1969), ICA-LiNGAM (Shimizu et al., 2006). Considering the huge sample imbalance in the test set, we use AUROC to evaluate how these methods work. To calculate the AUROC metric we first calculate the receiver operator characteristic curve (the true positive rate as a function of the false positive rate). The integral of this curve is the AUROC metric. We test all methods on pairs of time series, just like when training our models.

We can thus compare the learned causal discovery and classical techniques (see Appendix for experiment details).

We use all the methods using data from the game Donkey Kong and will only use other games for testing procedures (see Table 1). The learning approaches based on LSTM, TCN and Transformer each achieve above 90% AUC, while the traditional methods only achieve up to 60%. Using LSTM as the sequence encoder gives us the best performance 94%, and our procedure shows stable high inference quality even when changing the core encoder. As a semantically similar concept to causality and a common tool to explore connectivity, correlation here achieves only 59% AUC. Clearly, learned causal discovery far outperforms the methods based on human intuition.

Table 1: Methods comparison on game Donkey Kong

| METHOD | AUROC |
|---|---|
| Pearson Correlation | $0.585 \pm 0.031$ |
| Mutual Information | $0.569 \pm 0.030$ |
| Linear Granger Causality | $0.570 \pm 0.011$ |
| ICA-LiNGAM | $0.513 \pm 0.011$ |
| LSTM (ours) | $\mathbf{0.940 \pm 0.004}$ |
| TCN (ours) | $0.936 \pm 0.014$ |
| Transformer (ours) | $0.928 \pm 0.011$ |

### 3.3 Evaluation under Noise

In contrast to ideal microprocessor simulation, real-world causal discovery often happens in domains where there are all kinds of noise. For instance, the wire used to sample the voltages of transistors might be disrupted by the wind while recording brain waves. There is a lot of tool/observation noise other than noise in the system, collected with data together. Intending to explore the potential effectiveness of our methods in other real-world scenarios, we simulate the observation noise by simply adding Gaussian noise of different scales in both training and testing. Here we repeat the same steps we have done in Section 3.2. Our method still achieves above 90% AUC and outperforms traditional methods (see Figure 5). Our learning procedure is robust and only lightly affected by noise, while the performance of the conceptual methods significantly degrades below 50%. This benefits from the introduced supervised signal, which, arguably, makes our model regard noise as data augmentation while methods based on human intuition failed to extract effective features.

### 3.4 Internal Invariance

Linking cause and effect, causality should be stable in different environments wherever the data distribution of the environment changes or not, empowering its more robust transferability and generalization on internal than correlation. We thus take the other two games, Pitfall and Space Invaders as different behaviors from Donkey Kong but executed by the same complex system, to evaluate if our causal discovery approach is invariant to domain shifts. We are slightly concerned that our algorithm may work by memorizing aspects of the time series. Therefore, to reduce such data leakage, we only use the unique transistors in the other two games to construct test sets. We directly use the model trained on Donkey Kong to infer the causal relationships in Pitfall and Space Invaders and calculate their AUROC (see Figure 6). We find very strong generalizations across games. The learning of causal discovery leads to the causal discovery that is robust across games, which could be the result of our model having learned the causal dynamics among transistors.

### 3.5 Test if the methods also work in a different domain of causal discovery, fMRI data

Sudden and gradual changes both happen in real-world scenarios. While experiments in MOS 6502 focus on the sudden changes in digital voltage, we here examine our procedure in NetSim (Smith et al., 2011), a synthesized fMRI dataset modeling gradual changes in brain networks. There are in total 28 simulations generated with different specified system properties such as external input strength and the number of nodes in the system. Each simulation has a fixed number of nodes $N$ and specified property in all samples, where a sample refers to $N$ state sequences of $N$ nodes and its corresponding $N \times N$ connection matrix. It is a classical causal discovery setting allowing us to compare the multi-variate and graph-based discovery methods with us.

Methods Comparison on DonkeyKong

Figure 5: Methods comparison in the context of noise. Trained with different scales of noise, our method is evaluated with the conceptual methods together on the multiple noisy test sets.

We use all 18 simulations that have the same sequence length (200), and divide all samples in each simulation into the training set and test set in a ratio of 6 : 4. We omit perturbations here (we have the full ground truth connectivity matrix after all) and directly convert all the connection strengths into a binary coding of connections so that they could be represented as causal relationships. Note that here we only use the training set from simulation $1, 2, 3, 4, 8$ but the test set of all simulations, to both evaluate our procedure's effectiveness on similar and different scenarios. We merge all the training sets from these five simulations and all the test sets as the total training set and test set in our learning procedure and report the average AUROC in all methods. Here we only examine our procedure with the Transformer as the sequence encoder and learn one neural network estimator to deal with different simulations. We test traditional methods including ICA-LiNGAM, GES (Chickering, 2002), Mutual Information and Linear Granger Causality, and another supervised learning causal discovery method SLDisco (Petersen et al., 2022). Simulation 4 is a large system than the others with node number 50, which takes a lot of time for multivariate methods such as GES and Multivariate Granger Causality to converge, therefore we do not report them here. We see that in Figure 7.A, our procedure can capture effective causal features inside systems and overall outperforms the other methods. What's more, as reported by Löwe et al. (2022), even though Amortized Causal Discovery aims at capturing the shared dynamics among different underlying causal graphs, it achieves only $0.688 \pm 0.051$ on simulation 3. It shows the importance of learning causality with perturbation information. However, our procedure is suboptimal on the simulation 22, since its system property is nonstationary connection strengths and the training data might be too small for the neural network to learn such complex dynamics. However, in most cases, learned causal discovery works very well (see Appendix for inferred adjacency matrix examples).

Although noise propagates through the causal chain in a potentially understandable way, it is inevitable that the data collection faces some noise outside the system, such as errors caused by observational tools. Here we simulate such noise as we have done in the noise simulation in subsection 3.3 to see how methods work in

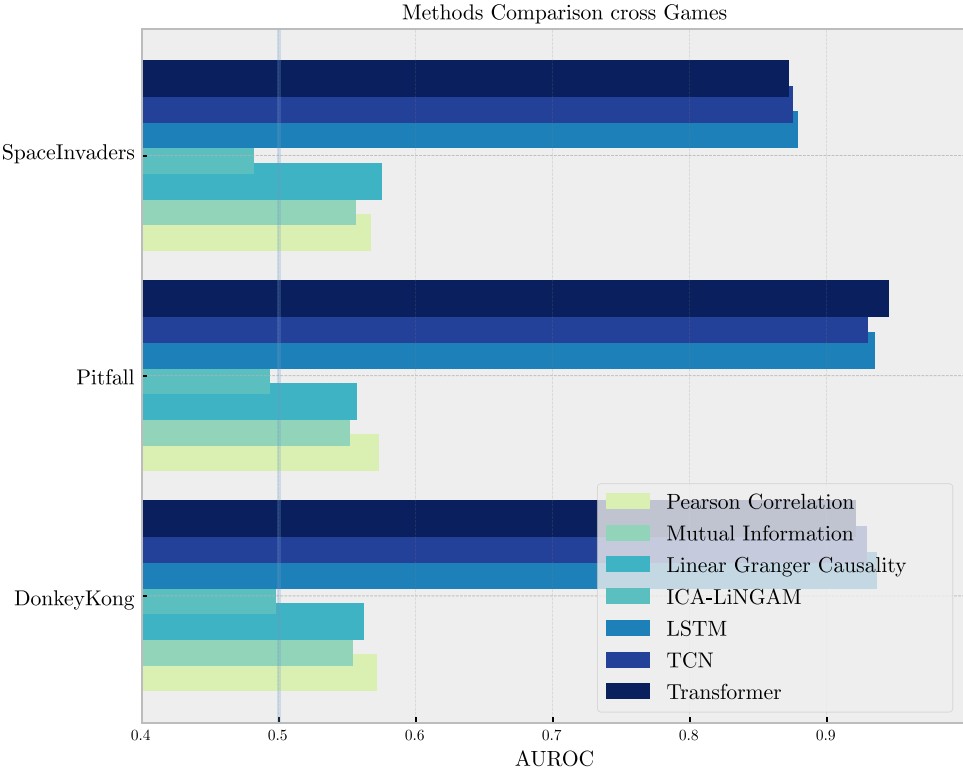

Figure 6: Methods comparison on three games. Our models (LSTM, TCN, Transformer) are only trained on Donkey Kong and tested on three games. Classical methods are directly tested on three games.

more realistic scenarios. Note that here we do not retrain the network as in the MOS 6502, we attempt to directly test the network we get from the noise-free training set on the test set with additive noise. Adding noise with a scale of 0.5 can severely affect the inference quality of traditional methods, while our procedure presents consistent anti-noise ability as it does in the MOS 6402 (Figure 7.B). Learned causal discovery is remarkably noise tolerant.

### 3.6 Explanation Study

We see that our learning procedure has shown good performance in causal discovery, but what if it is exploiting some shortcut that we humans do not understand and that is specific to the MOS 6502 or NetSim? To find out what our models have learned from the supervised signal, we use the model with the transformer to conduct an explanation study. We first use Gradient-weighted Class Activation Mapping (Grad-CAM) (Selvaraju et al., 2016) to try to understand what models were learned. Then we try to inspect what happens when we violate the temporal order of cause-result by adding a small time shift to the cause transistor.

Grad-CAM (Selvaraju et al., 2016) is a robust tool making Convolutional Neural Networks more transparent by helping localize saliency in the feature map. We adopt Grad-CAM mainly on the output of the attention block in the last layer and interpolate the feature map back to the same length of the input sequence. We see that in the first row of Figure 8, the model trained on regular recording shows high saliency at the windows where cause and effect have interactions. The high saliency here indicates that the model captures highly relevant features to recognize a causal relationship in the system used to supervise, such as the time lag of cause-effect. In the second row, we show the feature map of the model trained at a scale of 0.5. Similar to the model in the ideal context, the feature map in the noise model exhibits special attention at the windows cause-effect happens - as such the behavior of the model does make sense.

### A. Methods comparison without noise

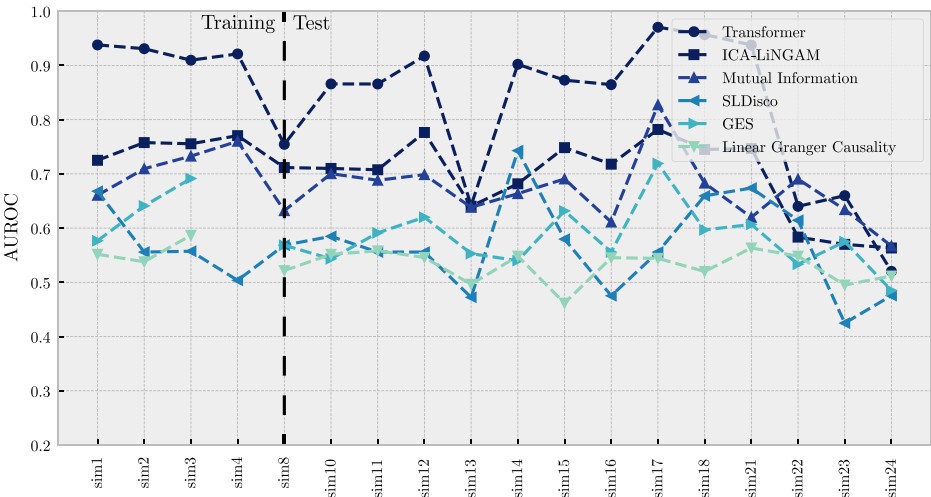

### B. Methods comparison with noise

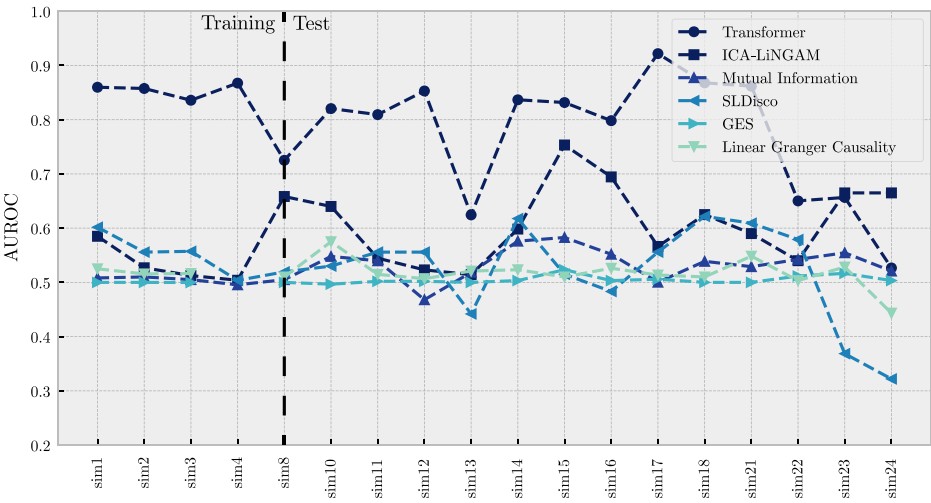

Figure 7: Methods comparison on the test set of 18 simulations in Netsim, each depicting a different causal system, in the context of noise-free and noise scale of 0.5 (only 60% samples on the left side of black vertical dotted line is used to train).

The precedence of cause is a core rule used to discriminate between cause and effect. Inspired by the time lag between cause and effect, we expect our model to give a different answer when it receives a modified input that violates the precedence of cause. Therefore, we move the effect transistor backward 200 time steps and infer the relationship between this new pair. As shown in Figure 9, when the effect precedes the cause, the model rejects the previously recognized causal relationship. We suggest that our model understands the basic rule between cause and effect and has adopted it to do causal discovery since the rejection decision is consistent with the broken causal chain (see Appendix for more examples).

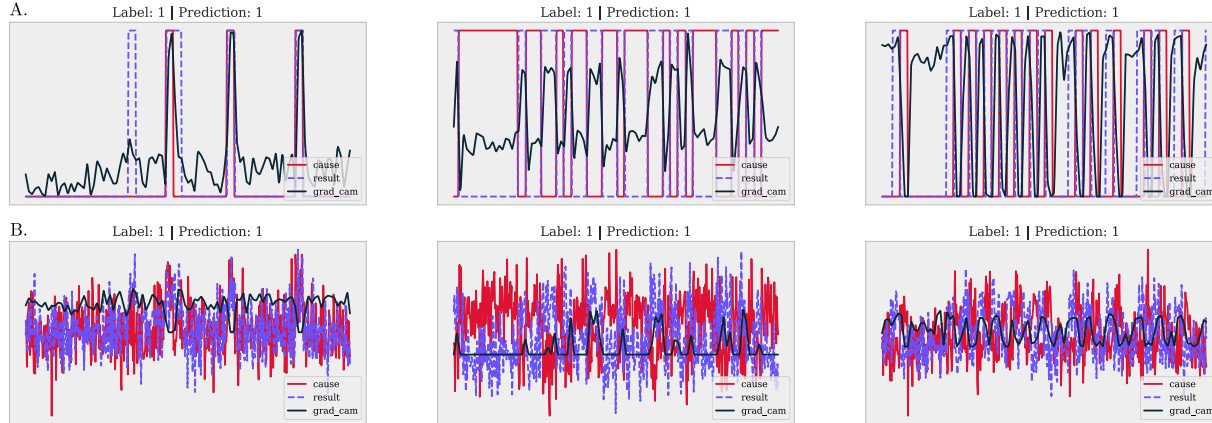

Figure 8: Grad-CAM mappings of causal pairs. **A.** The gradient saliency mapping of three causal transistor pairs in the ideal context. **B.** The gradient saliency mapping of the same causal transistor pairs at a noise scale of 0.5.

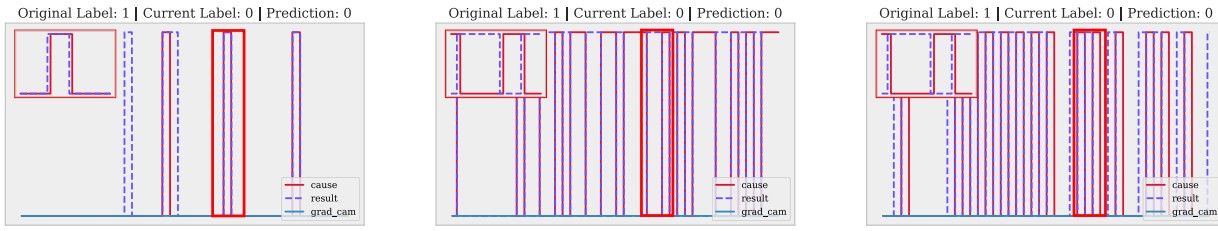

Figure 9: Cause-effect temporal reversal by making effect precedes cause.

## 4 Discussion

Here we have shown that causal discovery from observational time-varying data can be phrased as a learning problem. We use a causal system with many causal relations, the MOS 6502 for our main experiments. We find that the learned strategy far outperformed traditional human-designed methods, in particular in the high-noise domain. We find that learned causal discovery generalizes across different games. Lastly, we observe that the learned strategy, as should be expected, focuses on times where state transitions unfold.

The right causal discovery strategy will undoubtedly, differ across domains. We test our methods on microprocessor causality and synthesized brain networks. We acknowledge that a method that works well on the microprocessor may not work well in other real-world scenarios in such as medicine or neuroscience. Nevertheless, our focus on the microprocessor simply comes from a lack of data in other domains: neither medicine nor neuroscience have large datasets of observational data along with ground truth perturbation-based causality data. As these datasets become available, re-visiting learning approaches to causality will become a crucial issue.

The right causal strategy will also depend on noise and data quality. Real-world data contains noise, and often observations are only partial. Observations are also often short. The evolving internal state of other systems also produces latent noise that will be hard to learn. How to make causal discovery strategies noise tolerant and work within the limit of relatively small datasets is an important issue for future data.

Here we use perturbation analysis to define ground-truth causality. However, these causal effects will still generally depend on the network state and we certainly do have not enough data to cover all relevant

network states. Here the perturbation quality is vital to our supervised signal. Future developments should at short-term perturbations and develop perturbation analysis methods for more noisy real-world systems.

Our results are superior to the ones based on human assumptions (AUC .9 vs humans at .6). Furthermore, the results show a certain generalization ability even in a more limited observational setting, and the potential for transferability in the homogenous system. It is truly a limitation that our learned estimator can only have a local view of a pair of elements in a system, which makes it hard to acquire a more broad perspective like other multi-variate and graph-based causal discovery methods. However, because of the pair-wise input form, our estimator can do inference on arbitrary systems that have a different number of nodes, which allows it to conveniently transfer learned strategy to other systems, even potentially to other domains. As such, it seems clear that the advantage of learning is significant in our system. Learning may thus make a big difference relative to the current state of the art in many fields like public policy or epidemiology.

Currently, algorithms for causal discovery from observational data are constructed using mathematical ideas. For example, the Hyvarinen approach is based on sparseness (Shimizu et al., 2006), and the Blei deconfounder is based on independent confounders (Wang & Blei, 2019). However, it is quite unclear how good these assumptions are. Our approach can in principle discover ideas like deconfounders or sparse noise, but, importantly, it can use all of these ideas, those expressed by clever mathematicians and those that no one has yet discovered. Learning causal discovery promises to make the field more efficient.

The success of learning causal discovery which we present here suggests that we should use such approaches across the sciences. We advocate for large projects in the domains of public policy and epidemiology to produce datasets of ground-truth causality. A lack of proper benchmarks has long been holding back these fields. Our paper adds an extra reason for why we should produce such datasets: it promises to considerably improve the causal discovery procedures that power these fields.

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

# A    Appendix

## A.1    Experiment Details

In the MOS 6502 experiments, we use Pearson Correlation, Mutual Information, ICA-LiNGAM, Linear Granger Causality, and our procedure. For correlation, we calculate the AUROC on the absolute value of the Pearson correlation (implementation from Sklearn (Buitinck et al., 2013)). Not taking absolute value gives a little bit worse performance. For the Mutual Information, we take the implementation from Wu et al. (2020) which optimizes for float value input, and calculate AUROC on it. For the ICA-LiNGAM, we take the implementations from Causal-learn (https://github.com/cmu-phil/causal-learn). For the Linear

Granger Causality, we modify the implementations in Causal-learn and regard the maximum non-linear weight of LASSO regression result across time lags as the inferred causality, and calculate the AUROC on it. For our procedure, we calculate the AUROC directly on the network's output probability. All the operations mentioned above are done in a pair-wise way.

In the NetSim experiments, we use Mutual Information, ICA-LiNGAM, Linear Granger Causality, GES, SLDisco, and our procedure with Transformer as sequence encoder. For Mutual Information and our procedure, it is the same as we have done in the MOS 6502 experiments. For ICA-LiNGAM, GES, and Linear Granger Causality, we use similar implementations as we use in the MOS 6502 experiments but in a multi-variate regression way (see Causal-learn). We implement the SLDisco as the paper mentioned by ourselves. We calculate the AUROC on all of outputs of these methods. Only our procedure and Mutual Information are done in a pair-wise way.

In the explanation study, we modified the implementations of Grad-CAM from Gildenblat & contributors (2021) to align with the temporal sequences instead of images.

## A.2 Supplement Materials of Grad-CAM and Temporal Reversal

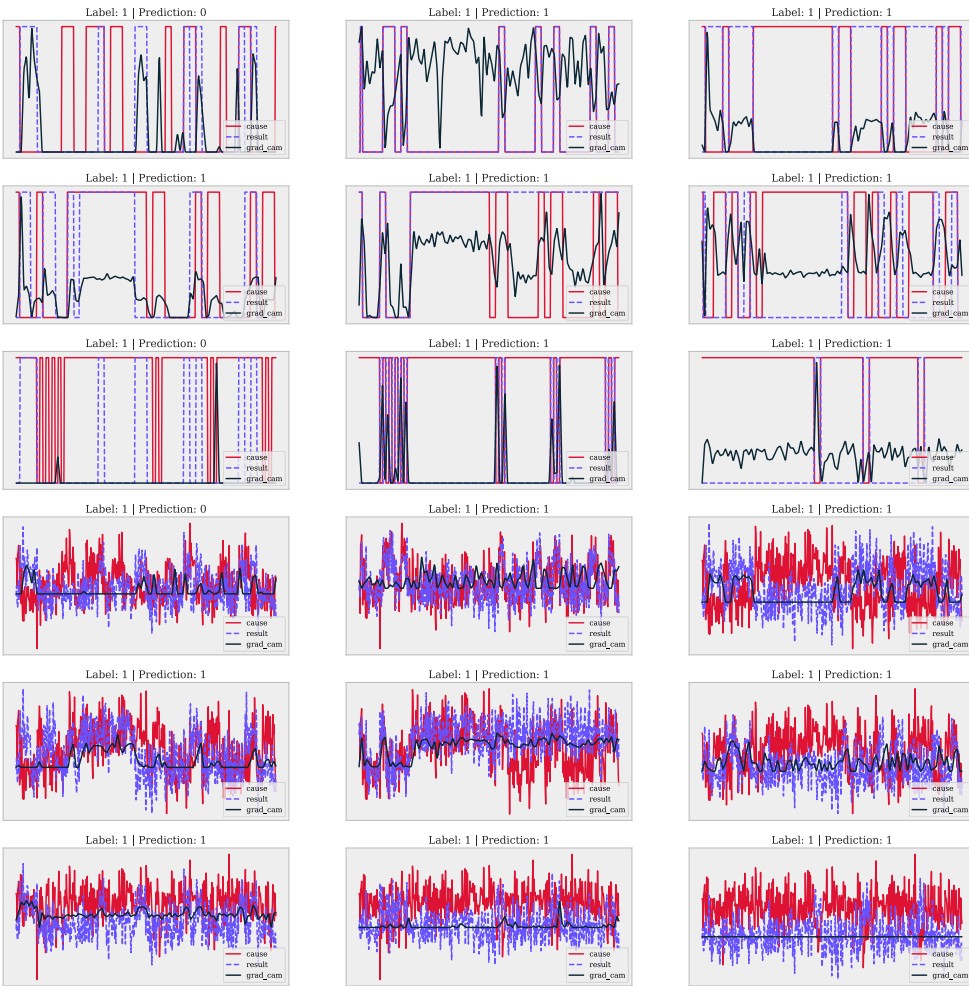

Figure 10: Grad-CAM supplement in MOS 6502 for some other samples (only shows the positive gradient saliency)

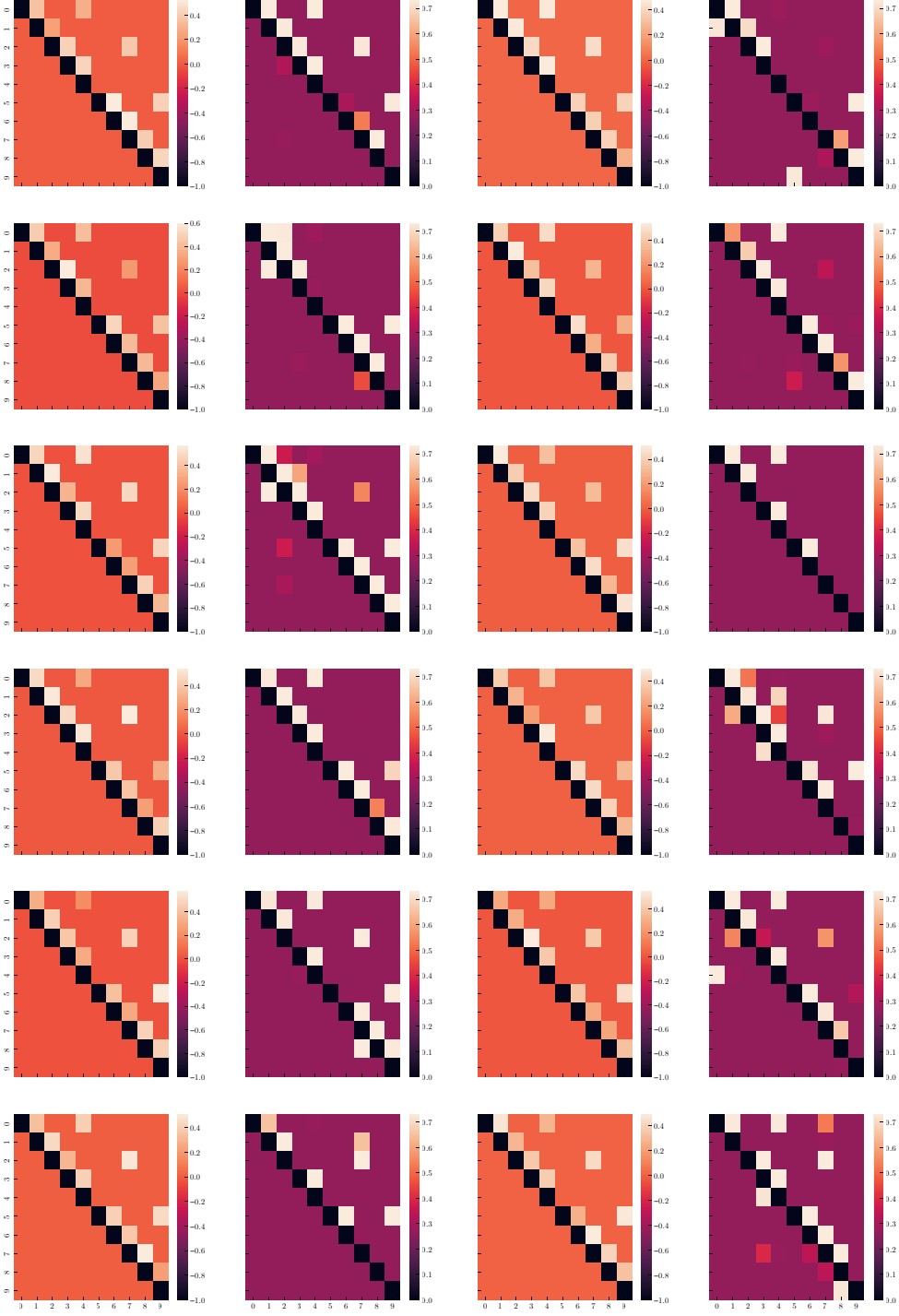

Figure 11: Ground truth adjacency matrix and inferred adjacency matrix on Simulation 17 in NetSim (Column $0, 2$ shows raw ground truth connection strength matrix, column $1, 3$ shows inferred matrix shows the inferred $P(Causal|\boldsymbol{X})$)

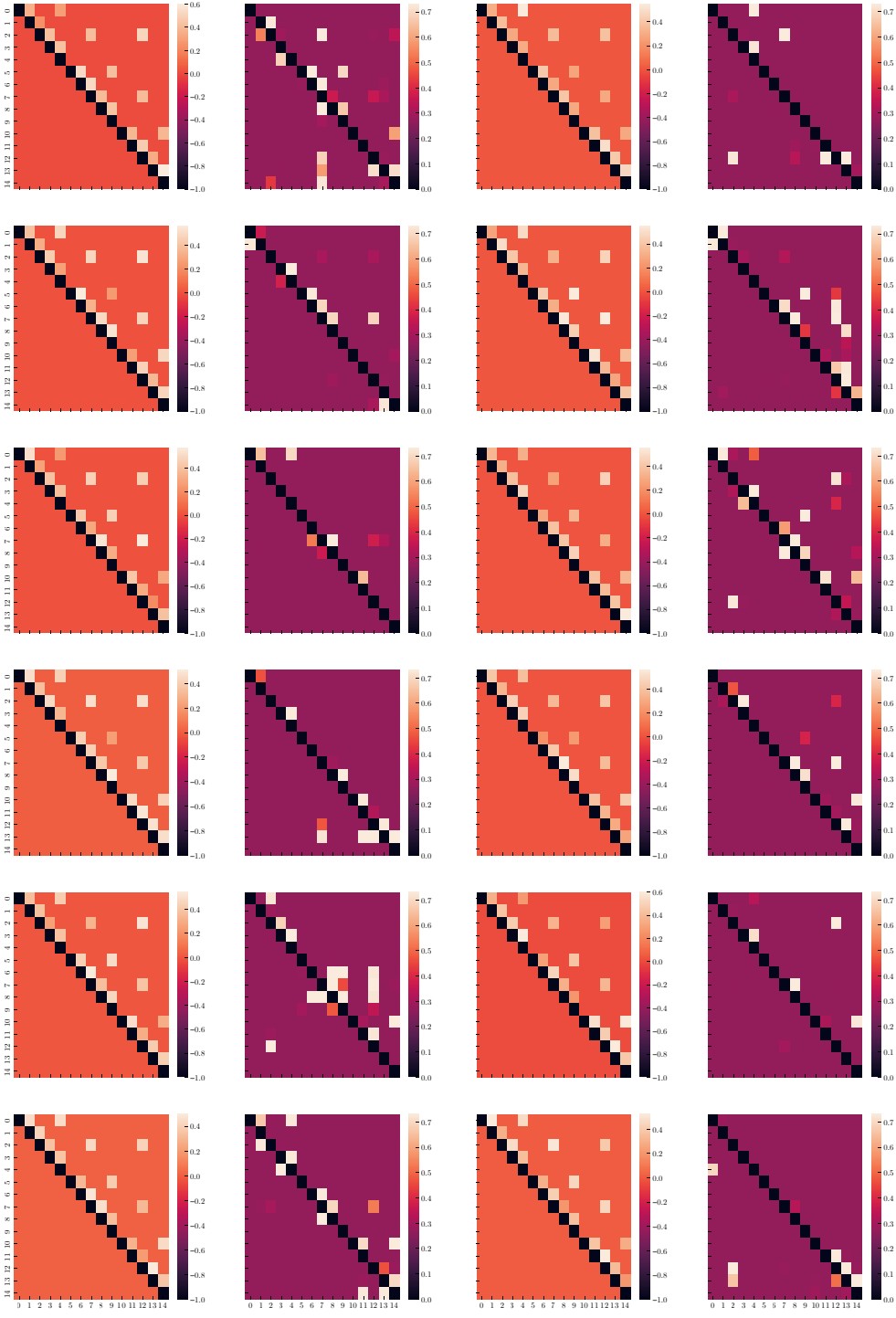

Figure 12: Ground truth adjacency matrix and inferred adjacency matrix on Simulation 3 in NetSim (Column $0, 2$ shows raw ground truth connection strength matrix, column $1, 3$ inferred matrix shows the inferred $P(Causal|\boldsymbol{X})$

