# OpenReview forum: "Learning Causal Discovery"
_TMLR — Rejected by TMLR_

### Review · Reviewer_RcXC · 2023-02-15

**Summary Of Contributions:**

The paper describes how machine learning could be used for causal discovery instead of pre-defined rules or metrics (based on "human intuition"). This is an interesting perspective. In an offline digital system, it is easy to use the proposed approach to learn to recognize if two signals are causally related. The method performs much better than existing rules measure/heuristics and can handle noise well too.

**Audience:**

Yes

**Broader Impact Concerns:**

"We advocate for large projects in the domains of public policy and epidemiology to produce datasets of ground-truth causality." It is not clear that large-scale perturbations are reasonable in such systems. More discussion is needed.


**Claims And Evidence:**

Yes

**Requested Changes:**

Clarify the pair embedding into a sequence of window embeddings.
Clarify the jargon regarding cycles and half cycles.
Comment more fully on how this could be used in practice. Perhaps via active learning through an optimized sequence of perturbations?
This would require additional results in small sample cases (ideally in more diversity of cases, say from control).

Instead of only finding unique patterns, it may be interesting to use a distance metric and remove all neighbors of a certain radius to remove redundant and  nearly redundant patterns.

**Strengths And Weaknesses:**

Strength:
The idea of using supervised learning to perform causal discovery is interesting. Causality is difficult to test from observational data, and the proposal could be used in a sort of active learning where new causal connections are predicted.

The idea of using an emulated microprocessor is also an interesting testbed.

The idea to interpret the results using Grad-DAM seem meaningful (in the low noise setting).

Weaknesses:

*There are some unclear steps:*

The conversion of the paired signals "into a sequence of window embeddings" is not clear.

The discussion of the netlist and half clock is unclear to someone from the machine learning community.

" 5 times with different seeds to guarantee the fairness of our results." I don't think this guarantees fairness.

*Practical significance is weak:*

The threshold at 0 for the average treatment effect would seem to be sensitive to noise (more false positives). Wouldn't larger treatment effects be important to learn from? This seems to be verified by the drop in AUCROC for noise in the second result.

The type of generalization tested is very weak because it is the same computer system, operating in very similar modes, even though the software is different. "We find that learned causal discovery generalizes across different games. " However, as noted the system may be memorizing redundant (or nearly redundant patterns).

The type of intervention test is performed in a digital emulation of a microprocessor, allowing the gathering of large amounts of data. In a physical microprocessor it would be difficult to intervene in the same manner to physically (electrically) manipulate the system. This method seems limited to digital systems that can be probed offline.

Together the weaknesses indicate that the approach is limited to a sort of memorization or interpolation of the pattern pairs that are causally related.

---

### Review · Reviewer_e7A9 · 2023-02-23

**Summary Of Contributions:**

The authors consider causal discovery by learning as inferred using deep learning approaches considering three model specifications; long-short-term-memory units (LSTM), convolutional networks and transformers respectively to identify causal relations in time-series data. Their approach is contrasted simple correlation and mutual information based procedures as well as standard (linear) Granger causality and ICA-LinGAM finding the proposed approach substantially outperforming these methods on datasets of transistor interactions in the MOS 6502 processor when executing programmes on games. Notably, the procedure relies on supervised training to identify the structure of the causal graph by inferring causal structures from sequence encoded embeddings considering LSTM, temporal convolutional network (TCN) and Transformers. They further find their procedure to be robust to noise and variation in the causal graph when trained on one game and tested on others. They further examine their procedure on simulated fMRI based on the NetSim toolbox also including SLDisco in the comparison. Model explainability is further considered by use of the Grad-CAM procedure.

**Audience:**

Yes

**Broader Impact Concerns:**

The broader impact is not in general discussed. Arguable the authors consider causal discovery in fairly harmless domains, however, discussions on limitations of such inferences, consequences of wrong inferences based on such modeling approaches as well as the importance of causal discovery as enabled by the proposed approach also for such discovery in larger systems would be good to further elaborate upon.

**Claims And Evidence:**

Yes

**Requested Changes:**

The authors should make available code and datasets which will strengthen the impact of the paper, enabling further research in the interesting datasets generated on transistor causal dynamics, as well as enhance reproducibility and transparency of their research, which I consider a must.

The authors need to further review the existing literature on causal discovery. It is fine to limit this review to only causal discovery for time-series data, however, even this literature is vast and not sufficiently covered by the authors. In particular, the authors only compare to very simple baselines (mutual information, correlation, linear granger causality and ICA-LINGAM), however, there are a larger bode of literature also exploring deep learning approaches which is not considered by the authors or even reviewed. This includes:
* Peters, Jonas, Dominik Janzing, and Bernhard Schölkopf. "Causal inference on time series using restricted structural equation models." Advances in neural information processing systems 26 (2013).
* Shajarisales, Naji, et al. "Telling cause from effect in deterministic linear dynamical systems." International Conference on Machine Learning. PMLR, 2015.
See further the extensive review of causal methodologies for time-series analysis encompassing also structural equation models as well as providing a review of relevant existing deep learning based causal discovery methods (see 4.2.3 Deep learning-based methods):
**Moraffah, Raha, et al. "Causal inference for time series analysis: Problems, methods and evaluation." Knowledge and Information Systems 63 (2021): 3041-3085.

Additional deep learning based causal discovery methodologies include methodologies such as:
* Ahmad, Wasim, Maha Shadaydeh, and Joachim Denzler. "Causal inference in non-linear time-series using deep networks and knockoff counterfactuals." 2021 20th IEEE International Conference on Machine Learning and Applications (ICMLA). IEEE, 2021.

* Peng, Wei. "Dli: A deep learning-based granger causality inference." Complexity 2020 (2020): 1-6.
Whata, Albert, and Charles Chimedza. "Evaluating uses of deep learning methods for causal inference." IEEE Access 10 (2022): 2813-2827.

The authors need to more carefully position and contrast their work to relevant parts of this larger body of related literature.

Minor comments:
There are several benchmark datasets for causal discovery in time-series - would be interesting if possible to furhter investigate the proposed approach on these, see for instance: https://causeme.uv.es/

The paper needs a bit of proof-reading:
human-assumption methods to exact. => human-assumption methods to extract.
empowering its more robust transferability and generalization on internal than correlation => What is mean by “on internal”?
“It is a classical causal discovery setting allowing us to compare the multi-variate and graph-based discovery methods with us.” => What is meant by “with us”

In summary, I find that the paper needs to be substantially improved considering the above.

**Strengths And Weaknesses:**

Strengths:
•	The proposed use of deep learning for causal discovery is valid and interesting.
•	The defined causal graphs based on transistor networks form a to my understanding novel and interesting testing ground for evaluating methods for causal discovery based on time-series data of general utility.
•	The paper is evaluated considering generalization across games as well as in terms of robustness to noise well probing the developed procedures.

Weakness:
•	The method is very limited compared to current-state-of-the-art.
•	There are many benchmarks data for causal discovery and it is unclear why the developed procedure is not also evaluated on these benchmarks.
•	The paper can be improved in its presentation (see requested changes) - in particular in the introduction relating to the existing deep causal discovery literature and include further comparisons to the existing literature.
•	Code and data is not made available which would increase impact as well as transparency substantially.
•	It needs to be clarified what the reported error bars in Table 1 and Figure 5 is given for. Importantly, the authors need to assess performance across model initializations/runs of the training procedure and it is unclear if this has been assessed.
•	It is somewhat unclear why error bars are not provided for the results in Figure 6.

---

### Review · Reviewer_Euax · 2023-02-27

**Summary Of Contributions:**

This work addresses the problem of causal discovery from time series data using a supervised approach where first parts of the system are perturbed and the causal connections recorded. This forms a training set where a model is trained to discriminate causal variable pairs from non-causal pairs from their observational trajectories. This is then tested on a test set of variable pairs with causality determined in the same manor. This casts causal discovery from time series as a supervised task with the limitation of requiring known causal (and non-causal) variable pairs in the same system to learn from. This is different than the classic observational causal discovery setup where we only have observational data from the system.

Experiments are performed on two simulated systems, a transistor system (MOS 6502) and an FMRI model. A time series model is learned on 60% of the causal pairs to predict whether unseen pairs (from the same model with causality defined in the same manor) are causal. It is also shown that these results are robust to Gaussian noise on both models. The proposed method performs extremely well on these tasks (>0.9 AUC vs. ~0.6 AUC) as compared to unsupervised baselines.

**Audience:**

Yes

**Claims And Evidence:**

No

**Requested Changes:**

- Make the claims inline with the experiments. To support the current claim I believe substantially more experimental evidence that the proposed method generalizes learning on one system to causal discovery on new systems is required. Alternatively, I would be in support of substantially reducing the claim. I believe the current experiments support something to the effect of knowing some of the causal relationships in a system can allow us to guess the other causal relationships with high accuracy. This implies that  in causal discovery we might be able to examine a small part of a system then generalize to the rest of the system with something like the proposed method. This could be helpful in causal discovery in high dimensional systems. For this it might be interesting to only use known causal relationships for variables in part of the space (for example the lower right corner of the chip) and attempt to generalize to the rest of the system. This is critical.
- Other suggestions above I regard as more minor and would improve the work but are not necessary in my view.

**Strengths And Weaknesses:**

Strengths:

- Improving algorithms for causal discovery from time series data is an interesting problem.
- Empirically establishes that causal pairs can be predicted from known causal pairs, at least in an 80/20 iid split, across two systems with ground truth causality defined based on single perturbations.
- The presentation makes the setup of the experiments quite clear.

Weaknesses:

- My main concern with this work is that the claim, especially as implied in the title “Learning Causal Discovery”, is not supported by the empirical evidence presented. Namely, the procedure tested relies a great deal on a large number of known causal and non-causal variable pairs in the system of interest, which, as the authors acknowledge is very rare in practice. Comparing this to methods such as Pearson Correlation and Granger Causality, which do not utilize such known causal information is not adequately supportive of this claim in my view. TMLR poses two questions to submissions, the first being “Are the claims made in the submission supported by accurate, convincing and clear evidence?” I believe the claim to be learning a causal discovery algorithm must be supported by applying this algorithm to systems where it is not given information on ground truth causal and non-causal pairs. For example, an experiment where a model trained on MOS 6502 was able to generalize on FMRI data.
- This model examines a very specific causal relationship. A clearer definition / examination of what variables are causal and which are not under this model would improve this presentation. If I was to summarize, section 2.1 defines causality in the MOS system as: Variable i causes variable j if and only if when i is replaced with “not i” in isolation on the simulation of a specific program, the sequence of variable j changes within 256 half-clocks. This ignores limited perturbations (i only perturbed for a small amount of time), effects which can only be seen with multiple perturbations, and effects which take longer than 256 half clocks to register. Specifically, this is **not** the same notion of causality assumed in other algorithms (for example Granger Causality). I do believe this definition is useful, nevertheless it might benefit from some additional discussion and justification.
- “We test our methods on microprocessor casualty and synthesized brain networks” because ”Neither medicine nor neuroscience have large datasets of observational data along with ground truth perturbation-based causality data.” — Its definitely possible to get simulated data that is significantly closer to medicine and neuroscience than microprocessors. I would suggest rewording this.
- Why are we concerned with performance under Gaussian noise? In a binary system Gaussian noise is somewhat strange to consider.
- “Considering the huge sample imbalance in the test set, we use AUROC to evaluate how these methods work” — AUROC is notoriously poor for evaluation under imbalanced test data. AUPRC would be more useful to consider if this is a concern. Although given the current results this does not seem to be much of an issue to convey the point.

Other comments:

- Shouldn’t there be an absolute value in equation 1? Personally I would consider a different name for this as average treatment effect generally refers to averages across populations instead of average over time for a specific individual.
- Figure 3 seems to imply that some transistors have many more effects than others. What does this distribution look like? And what effect does this distribution have on the empirical results?

---

### Comment · Action_Editors · 2023-03-27
**One more week to complete the manuscript update**

Dear authors,
Following your request, we propose to grant you another week to complete the manuscript update.
I hope that this will be OK.
Sincerely,
Bertrand Thirion

---

### Decision · Action_Editors · 2023-04-12

**Recommendation:** Reject

**Comment:**

The authors need more time to work on their experiments.
We believe that there is a lot of value in this work. The way TMLR handles the situation is to reject for the moment, with an implicit recommendation to re-submit when the paper is ready.
Thank you for your contribution.

**Audience:**

TMLR's audience is certainly interested in the findings of this paper.

**Claims And Evidence:**

The authors are still working on the claims following the reviewer's feedback.